# Synergistic Effect of Clinically Available Beta-Lactamase Inhibitors Combined with Cefiderocol against Carbapenemase-Producing Gram-Negative Organisms

**DOI:** 10.3390/antibiotics11121681

**Published:** 2022-11-22

**Authors:** Gabriele Bianco, Paolo Gaibani, Sara Comini, Matteo Boattini, Giuliana Banche, Cristina Costa, Rossana Cavallo, Patrice Nordmann

**Affiliations:** 1Microbiology and Virology Unit, University Hospital Città della Salute e della Scienza di Torino, 10126 Turin, Italy; 2Operative Unit of Microbiology, IRCCS Azienda Ospedaliero-Universitaria di Bologna, 40138 Bologna, Italy; 3Department of Public Health and Paediatrics, University of Torino, 10124 Turin, Italy; 4Medical and Molecular Microbiology, Faculty of Science and Medicine, University of Fribourg, 1700 Fribourg, Switzerland; 5Swiss National Reference Center for Emerging Antibiotic Resistance (NARA), University of Fribourg, 1700 Fribourg, Switzerland; 6INSERM European Unit (IAME), University of Fribourg, 1700 Fribourg, Switzerland; 7Institute for Microbiology, University of Lausanne and University Hospital Centre, 1011 Lausanne, Switzerland

**Keywords:** cefiderocol, β-lactamase inhibitor, synergism, avibactam, vaborbactam, relebactam

## Abstract

The role of β-lactamases in reduced susceptibility or resistance to cefiderocol has been supported by recent reports. The purpose of this study was to investigate the *in vitro* impact of clinically available β-lactamase inhibitors on cefiderocol activity against characterized carbapenemase-producing Gram-negative isolates. A collection of 39 well-characterized Gram-negative isolates obtained from various clinical sources and countries were included. Cefiderocol antimicrobial susceptibility was evaluated via reference broth microdilution. The chequerboard microdilution method and time–kill assays were used to determine the synergy of tazobactam, avibactam, vaborbactam and relebactam in combination with cefiderocol. MICs of cefiderocol presented a 4- to 256-fold reduction against *Klebsiella pneumoniae* carbapenemase (KPC)-producing Gram-negative isolates (predominantly *K. pneumoniae*) when avibactam, vaborbactam and relebactam were combined individually. Notably, the KPC-inhibitors led to a 4- to 32-fold reduction in cefiderocol MICs in the four cefiderocol-resistant KPC-producing *K. pneumoniae* isolates, showing restoration of cefiderocol susceptibility (MIC ≤ 2 mg/L) in ten out of twelve cases. Tazobactam led to a 4- to 64-fold decrease in cefiderocol MICs only in *K. pneumoniae* strains harbouring *bla*_KPC-41_, *bla*_KPC-31_, *bla*_KPC-53_ and *bla*_KPC-66_. The synergistic effect of all serine-β-lactamase inhibitors on cefiderocol activity was also shown in OXA-48-like-producing Enterobacterales strains. Conversely, a combination of β-lactamases inhibitors with cefiderocol was not synergistic with all OXA-23-like-producing strains and most metallo-β-lactamases producers. In conclusion, the addition of clinically available serine β-lactamase inhibitors to cefiderocol might represent an important development in the formulation to increase its spectrum and therapeutic efficacy, and to limit *in vivo* resistance emergence.

## 1. Introduction

Cefiderocol is a novel siderophore cephalosporin with broad activity against Gram-negative bacteria. It is structurally similar to cefepime (pyrrolidium group on the C-3 side chain) and ceftazidime (carboxypropyl-oxymino group on the C-7 side chain), a characteristic that improves both stability against β-lactamases and transport across the bacterial outer membrane [1]. Furthermore, the chlorocatechol group on the end of the C-3 side chain confers siderophore activity and enhances hydrolytic stability against various β-lactamases, including carbapenemases. Further, binding to extracellular free ferric ions allows cefiderocol to be transported across the outer membrane via the iron transport system of Gram-negative organisms, overcoming resistance mechanisms such as efflux pumps upregulation and porin channel mutations [1].

Cefiderocol has been evaluated in large international surveillance studies, revealing promising activity against multidrug-resistant Gram-negative isolates [2,3,4,5,6,7,8]. In the SIDERO-WT surveillance program, the percentage of cefiderocol-resistant isolates was 0.4%, 0.6% and 0.7% in 2014–2015, 2015–2016 and 2016–2017, respectively [3,6,7]. Isolates with MIC > 4 mg/L were predominantly *Acinetobacter baumannii* (78.9%) and Enterobacterales (17.4%). The excellent *in vitro* activity of cefiderocol was also observed among isolates resistant to carbapenems. The SIDERO-CR study showed that 96.1% of isolates (1801/1873) were susceptible to cefiderocol, with most resistant isolates among *A. baumannii* (*n* = 38) and Enterobacterales (*n* = 31) [8].

Based on the mechanism of cefiderocol, mutations affecting the iron transporter systems are associated with clinical resistance. Mutations in *piu*D; *pir*R, *pir*A and *piu*A; and *cir*A have been identified in *Pseudomonas aeruginosa*, *A. baumannii* and *Klebsiella pneumoniae* clinical isolates, respectively [9]. However, the role of β-lactamases in reduced susceptibility or resistance to cefiderocol has been supported by several recent reports [9]. The SIDERO-CR study’s molecular investigation of resistant isolates with MIC > 4 mg/L identified the most common isolates as PER-producing *A. baumannii* and NDM-producing Enterobacterales [6,8]. Beta-lactamase inhibitors, including avibactam, clavulanic acid and dipicolinic acid, decreased cefiderocol MICs against Gram-negative isolates with a cefiderocol MIC of 8 mg/L, suggesting that serine β-lactamases and metallo β-lactamases may play a role in cefiderocol resistance [10].

*In vivo* emergence of cross-resistance to cefiderocol following treatment with ceftazidime/avibactam or ceftolozane/tazobactam has also been described [9]. Emergence of cross-resistance to ceftazidime/avibactam and cefiderocol in two *Enterobacter hormaechei* strains due to expression of the A292_L293del AmpC variant was described in two cefepime-treated patients [11]. In addition, Simner et al. reported ≥4-fold increases in cefiderocol MICs in *P. aeruginosa* isolates following ceftolozane/tazobactam treatment [12]. *In vivo* selection of *Klebsiella pneumoniae* carbapenemase (KPC)-producing *K. pneumoniae* co-resistant to ceftazidime/avibactam and cefiderocol after ceftazidime/avibactam treatment was previously described [13], associated with the expression of KPC mutants. Combining cefiderocol with clinically available β-lactamase inhibitors could therefore be a way to enhance its activity and reduce the risk of *in vivo* emergence of resistant strains. *In vitro* and *in vivo* studies evaluating the activity of cefiderocol plus β-lactamase inhibitors are limited. The purpose of this study was to investigate the *in vitro* impact of four clinically available β-lactamase inhibitors on cefiderocol activity against diverse carbapenemase-producing Gram-negative isolates.

## 2. Results

### 2.1. Antimicrobial Susceptibility Testing and Chequerboard Assays

The MICs of tazobactam, vaborbactam and relebactam β-lactamase inhibitors for all 39 carbapenemase-producing strains were >64 mg/L, whereas the MICs of avibactam ranged from 16 mg/L to >64 mg/L, indicating that none of these molecules have intrinsic activity against those strains. Of the 39 strains, 30 were susceptible to cefiderocol (MICs range 0.03–2 mg/L). Nine strains were cefiderocol-resistant, of which four were KPC-producing *K. pneumoniae* (MICs range 4–16 mg/L), three were Enterobacterales NDM-producers (MICs range 4–8 mg/L), and two were *A. baumannii* NDM and OXA-23-like co-producers (MICs 8 mg/L and 16 mg/L, respectively) (Table 1). 

As shown in Table 1, avibactam, vaborbactam and relebactam combined with cefiderocol had a synergistic effect on all KPC producers, regardless of other β-lactamases co-expressed. MICs of cefiderocol presented a 4- to 256-fold reduction when avibactam, vaborbactam and relebactam were combined individually.

The synergistic effect of tazobactam was only observed on KPC-41-producing *K. pneumoniae* (N435), KPC-50-producing *K. pneumoniae* (N859), KPC-53-producing *K. pneumoniae* (CAZ59BO), and KPC-66-producing *K. pneumoniae* (KPC_TO3). Among the 11 metallo-β-lactamase producers, β-lactamase inhibitors combined with cefiderocol had a very low synergistic effect rate; FICI values < 0.5 were observed for combinations including avibactam, tazobactam or relebactam on only three strains co-expressing metallo-β-lactamases (NDM or VIM) and various other β-lactamases belonging to Ambler classes A, C and D. No synergistic effect was observed for all OXA-carbapenemase producing *A. baumannii* strains, including those co-producers of NDM enzymes. Conversely, all β-lactamase inhibitors combined with cefiderocol showed a synergistic effect on all OXA-48-like-producing *E. coli* or *K. pneumoniae*.

### 2.2. Time–Kill Assays

Changes in bacterial load from 0 to 24 h for cefiderocol alone and in combination with β-lactamase inhibitors were tested against five representative carbapenemase-producing isolates (Figure 1). Time–kill curves (TKC) showed that cefiderocol alone at 1× MIC determined a decrease in viable counts of bacterial cells within 6–8 h, whereas regrowth occurred in all isolates tested. The TKC further displayed that avibactam, vaborbactam and relebactam, but not tazobactam, combined with cefiderocol had a synergistic effect on KPC_TO1 (*bla*_KPC-3_) and N859 (*bla*_KPC-50_). Likewise, TKC analysis confirmed the indifferent effect of all β-lactamase inhibitors on cefiderocol activity against NDM and OXA-23 producers (i.e., N1700 (*bla*_NDM-1_) and N1183 (*bla*_OXA-23_)). Lastly, tazobactam and avibactam, but neither vaborbactam nor relebactam, combined with cefiderocol were synergistic against OXA-48 producer *K. pneumoniae* N1081 (*bla*_OXA-48_).

Overall, time–kill experiments showed synergistic bactericidal effects of combinations involving avibactam and relebactam, for both combinations in *K. pneumoniae* N859 (*bla*_KPC-50_) and cefiderocol plus avibactam in *K. pneumoniae* N1091 (*bla*_OXA-48_). In these cases, bacterial cell viability in the cefiderocol/β-lactamase inhibitor combination continuously decreased within 24 h of incubation.

## 3. Discussion

Resistance to β-lactamases activity together with the “Trojan horse” entry strategy into bacterial cells represent the strengths of the recently approved siderophore-cephalosporin cefiderocol. Although surveillance studies have reported potent activity with low MIC90 and MIC50 values, even on MDR isolates, reports of cefiderocol resistance are steadily increasing [11,12,13,14,15,16]. The combination of several mechanisms, including mutations affecting function of siderophore receptors and expression of certain β-lactamases, seems to be the main cause of resistance to cefiderocol [9].

Herein, we evaluated the *in vitro* impact of four different clinical β-lactamase inhibitors on the activity of cefiderocol towards characterized carbapenemase-producing Gram-negative bacterial isolates. MICs of cefiderocol presented a 4- to 256-fold reduction against KPC-producing Gram-negative isolates (predominantly *K. pneumoniae*) when avibactam, vaborbactam and relebactam were combined individually. Three of the four cefiderocol-resistant isolates expressed KPC variants associated with ceftazidime/avibactam resistance (KPC-41, KPC-50 and KPC-31), highlighting the cross-resistance phenomenon recently reported in the literature [13,17]. Notably, the KPC-inhibitors led to a 4- to 32-fold reduction in cefiderocol MICs in the four cefiderocol-resistant KPC-producing *K. pneumoniae* isolates showing restoration of susceptibility to cefiderocol (MIC ≤ 2 mg/L) in ten out of twelve cases. These results support previously reported evidence of the role of combinations including cefiderocol and serine-β-lactamase inhibitors, such as avibactam and vaborbactam, against cefiderocol susceptible or resistant KPC-producing Enterobacterales [13]. Although all KPC-producing isolates co-expressed other β-lactamases, the impact of tazobactam on cefiderocol MIC was not significant, except on isolates expressing KPC variants conferring both ceftazidime/avibactam resistance and the ESBL phenotype. Indeed, tazobactam led to a 4- to 64-fold decrease in cefiderocol MICs in *K. pneumoniae* isolates harboring *bla*_KPC-41_, *bla*_KPC-31_, *bla*_KPC-53_ and *bla*_KPC-66_. [18,19]. The role of broad spectrum β-lactamases (e.g., SHV-type, CTX-M-type, PER-type and CMY-type) in cefiderocol activity has already been shown by experiments on isogenic mutants [20,21]. In accordance with this, we also observed the synergistic activity of cefiderocol plus all four β-lactamase inhibitors in *K. pneumoniae* ATCC 700603 isolate (*bla*_SHV-18_). The synergistic effect of all serine-β-lactamase inhibitors on cefiderocol activity was also shown in OXA-48-like-producing Enterobacterales isolates (N1067, N1091 and N1085) via chequerboard assays. Although tazobactam, vaborbactam and relebactam have no inhibitory effect on OXA-48-like carbapenemase [22], their addition caused a 4- to 8-fold decrease in cefiderocol MICs. However, synergistic effect was confirmed via time–kill assay only for the combination including tazobactam. 

The combination of β-lactamase inhibitors with cefiderocol was not synergistic with all OXA-23-like-producing isolates and most metallo-β-lactamase producers. Synergistic effects were only observed with avibactam, tazobactam and relebactam on three strains expressing NDM-1 or VIM-2 metallo-β-lactamases together with various other β-lactamases of class A, C, or D. However, the combination of cefiderocol plus serine-β-lactamase inhibitors showed additive effects on most metallo-β-lactamase producers, emphasizing the contribution of multiple β-lactamases expression on the activity of cefiderocol in metallo-β-lactamase producers [23]. Kohira et al. observed a ≤2-fold cefiderocol MIC decrease with the addition of avibactam or dipicolinic acid (an *in vitro* inhibitor of metallo-β-lactamase) in NDM-1-producing *K. pneumoniae* isolates from a multi-national surveillance study (SIDERO-WT-2014) [18]. Interestingly, the addition of both avibactam and dipicolinic acid led to an 8- to 64-fold decrease in cefiderocol MIC. The same study showed the synergistic effect of avibactam plus cefiderocol against 28 cefiderocol-resistant (MICs ranging from 4 to >32 mg/L) *A. baumannii* isolates. However, most of the isolates were producers of PER-1, a broad-spectrum β-lactamase associated with significant hydrolytic activity on cefiderocol [20,24,25]. In contrast, herein, the VIM-producing *P. aeruginosa* strain N1215 carrying *bla*_PER-1_ showed a low MIC against cefiderocol (0.25 mg/L), and the combination of cefiderocol with β-lactamase inhibitors showed additive effects with tazobactam and avibactam, and synergistic effects only with relebactam. This might highlight the importance of the level of expression of β-lactamase genes in different bacterial species to cefiderocol resistance.

A further observation of our study is the lack of cefiderocol antibacterial activity in monotherapy at 1× MIC concentration in time–kill experiments. Cefiderocol acquired bactericidal activity only when combined with avibactam (in *K. pneumoniae* N859 (*bla*_KPC-50_) and *K. pneumoniae* N1091 (*bla*_OXA-48_)) and with relebactam (in *K. pneumoniae* N859 (*bla*_KPC-50_)). In agreement with our results, Ni et al. observed regrowth of *A. baumannii* after 6 h of treatment with 1× MIC cefiderocol monotherapy in time–kill assays, in isolates both susceptible and resistant to cefiderocol [26]. 

This represents the first *in vitro* evaluation of the impact of the four main clinically available β-lactamase inhibitors on cefiderocol activity against diverse carbapenemase-producing Gram-negative isolates. However, this study had some limitations. First, the number of isolates tested and β-lactamase variants included were limited. Second, characterization of mechanisms of resistance to cefiderocol other than β-lactamase production was not performed in detail for all isolates.

## 4. Materials and Methods

### 4.1. Bacterial Isolates

A collection of 39 well-characterized Gram-negative isolates obtained from various clinical sources and countries were included in this study (Table 1). Bacterial species included were *K. pneumoniae* (*n* = 19), *A. baumannii* (*n* = 7), *E. coli* (*n* = 6), *P. aeruginosa* (*n* = 5), *Citrobacter freundii* (*n* = 1) and *Enterobacter cloacae* (*n* = 1). All strains were carbapenemase-producers: Ambler class A (KPC-type (*n* = 18)), Ambler class B (NDM (*n* = 10), VIM-type (*n* = 2) and IMP-type (*n* = 1)), Ambler class D (OXA-23 (*n* = 5), OXA-48-like (*n* = 3), OXA-40 (*n* = 1) and OXA-58 (*n* = 1)). Reference strains *E. coli* ATCC 25922, *K. pneumoniae* ATCC 700603, *K. pneumoniae* ATCC BAA-2814 and *P. aeruginosa* ATCC 27853 were also included. 

### 4.2. Cefiderocol Susceptibility Testing

Cefiderocol susceptibility was evaluated via reference broth microdilution using iron depleted Mueller–Hinton broth (ID-MHB) according to EUCAST guidelines [27]. *Escherichia coli* ATCC 25922 and *P. aeruginosa* ATCC 27853 were used as quality control strains for each experimental sitting, checking that quality control results were within the specified ranges. Susceptibility data were interpreted according to current EUCAST clinical breakpoints [28]. 

### 4.3. Chequerboard Assay

The chequerboard microdilution panel method determined the synergy of tazobactam, avibactam, vaborbactam and relebactam in combination with cefiderocol. Experiments were performed in triplicate using ID-MHB. Concentration of inhibitors was set at 4 mg/L and the final concentration of cefiderocol ranged from 0.007 to 16 mg/L. The fractional inhibitory concentration (FIC) was calculated as previously described [29]. Briefly, MIC of drug A or B in combination/MIC of drug A or B alone; the FIC index (FICI) was determined by summing the FICs of drugs A and B. According to the minimum FICI, the results of combination tests were interpreted as synergistic (FICI ≤ 0.5), additive (FICI > 0.5 and ≤1), indifferent (FICI > 1 and ≤4), and antagonistic (FICI > 4).

### 4.4. Time–Kill Assay

Five strains were selected for subsequent time–kill assays: two strains producing class A carbapenemase (KPC_T01 (*bla*_KPC-3_) and N859 (*bla*_KPC-50_)), one strain producing class B carbapenemase (N1700 (*bla*_NDM-1_)) and two strains producing class D carbapenemase (N1183 (*bla*_OXA-23_) and N1091 (*bla*_OXA-48_)). Time–kill assays were performed in duplicate by inoculating 2.5 × 10^5^ cfu/mL of the test organism into 10 mL of ID-MHB. Cefiderocol and β-lactamase inhibitors were tested alone and in combination. Concentrations were tested at 1× MIC value for cefiderocol and at 4 mg/L for β-lactamase inhibitors. ID-MHB without any drug was set as positive control. Bacterial suspension was added to each experimental group and shaken at 37 °C at 200 rpm. The bacterial load (log_10_ cfu/mL) was determined at 2, 4, 8, 12 and 24 h by removing 100 µL of each suspension, serially diluting in sterile water and plating on Nutrient Agar ISO 21528 (Liofilchem^®^, Roseto degli Abruzzi, Italy). The lower limit of accurately quantifiable cfu using this method was 1 log_10_ of viable bacteria per mL. Synergy was defined as a reduction ≥2 log10 cfu/mL at 24 h for drugs in combination compared with the most active drug alone. No interaction was defined as a <2 log_10_ cfu/mL increase or decrease at 24 h for the drug combination in comparison with the most active antibiotic alone [30]. Bactericidal activity was defined as a >3 log_10_ cfu/mL reduction in the antibiotic combination treatment compared with the untreated control at the start of each assay from the original inocula, whereas bacteriostatic activity was defined as a <3 log_10_ cfu/mL decrease.

## 5. Conclusions

In conclusion, our results support the role of carbapenemases and other β-lactamases as sources of reduced susceptibility to cefiderocol, and consequently, the potential contribution of β-lactamase inhibitors for recovering its efficacy. The addition of clinically available serine β-lactamase inhibitors for cefiderocol might represent an important development in the formulation to increase its spectrum and therapeutic efficacy, and to limit *in vivo* emergence of resistance. Our results indicate that avibactam might be the best partner, as it had a broader spectrum of action and provided a bactericidal synergistic effect in combination with cefiderocol.

## Figures and Tables

**Figure 1 antibiotics-11-01681-f001:**
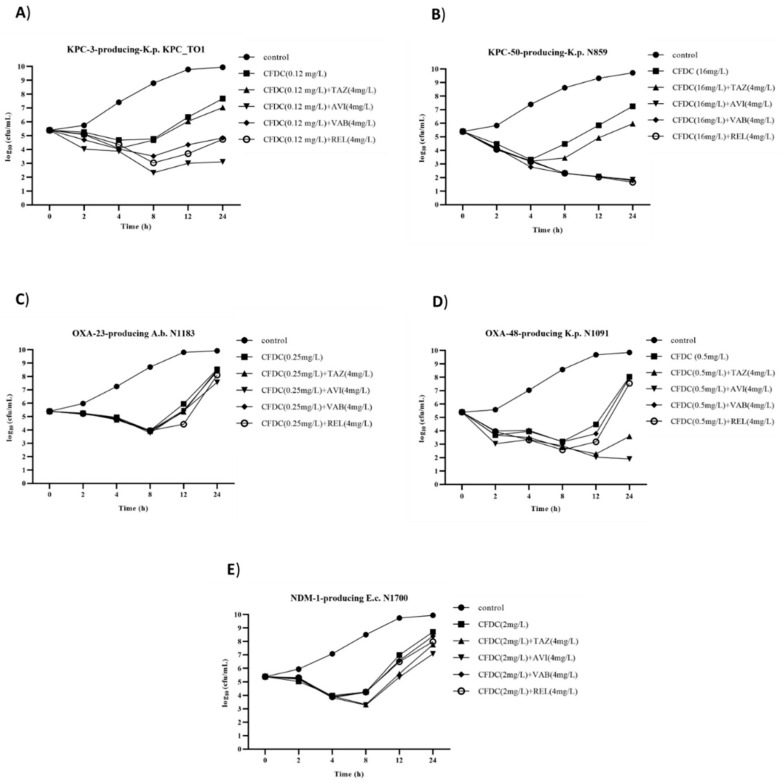
Time–kill curves of cefiderocol alone and combined with tazobactam, avibactam, vaborbactam and relebactam against KPC_T01 carrying *bla*_KPC-3_ (**A**), N859 carrying *bla*_KPC-50_ (**B**), N1183 carrying *bla*_OXA-23_ (**C**), N1091 carrying *bla*_OXA-48_ (**D**) and N1700 carrying *bla*_NDM-1_ (**E**). Data are presented as arithmetic means. Differences ≤ 0.5 log_10_ at each time point were observed. Time–kill curves of β-lactamase inhibitors alone against the five isolates tested overlapped with the positive control (differences ≤ 0.5 log_10_ at each time point) and were not included in the graphs.

**Table 1 antibiotics-11-01681-t001:** FICIs of tazobactam, avibactam, vaborbactam and relebactam in combination with cefiderocol against carbapenemase-producing Gram-negative strains.

Strain	Species	Sequence Typing	Carbapenemase Gene	Other β-Lactamases Genes	MIC (mg/L)					FICI and Interpretation			
					CFDC	CFDC+TAZ	CFDC+AVI	CFDC+VAB	CDFC+REL	CFDC+TAZ	CFDC+AVI	CFDC+VAB	CDFC+REL
Ambler class A													
BO318KP ^[a]^	*K. pneumoniae*	ST-512	*bla* _KPC-3_	*bla*_TEM-1_, *bla*_SHV-11_	4	2	1	1	1	0.56	**0.25**	**0.31**	**0.31**
N1118 ^[b]^	*K. pneumoniae*	-	*bla* _KPC-2_	*bla* _SHV-11_	0.12	0.12	≤0.007	0.015	0.015	1.06	**0.18**	**0.24**	**0.24**
N2350 ^[b]^	*K. pneumoniae*	-	*bla* _KPC-3_	*bla*_SHV-11_, *bla*_OXA-9_	2	2	≤0.007	0.25	0.125	1.06	**0.25**	**0.37**	**0.31**
CAZ156BO ^[a]^	*K. pneumoniae*	ST-101	*bla* _KPC-3_	*bla* _SHV-156_	1	0.5	0.125	0.06	0.125	0.56	**0.19**	**0.12**	**0.19**
BAT16KP ^[a]^	*K. pneumoniae*	ST-512	*bla* _KPC-3_	*bla*_TEM-1_, *bla*_SHV-11_	1	1	0.06	0.25	0.06	1.03	**0.12**	**0.31**	**0.12**
BAT15KP ^[a]^	*K. pneumoniae*	ST-512	*bla* _KPC-3_	*bla*_TEM-1_, *bla*_SHV-11_	1	1	0.12	0.25	0.12	1.06	**0.18**	**0.31**	**0.18**
KPC_TO5 ^[c]^	*K. pneumoniae*	ST-512	*bla* _KPC-3_	*bla*_TEM-1A_, *bla*_OXA-9_, *bla*_SHV-11_	0.5	0.25	0.015	0.12	0.015	0.56	**0.09**	**0.3**	**0.09**
KPC_TO1 ^[c]^	*K. pneumoniae*	ST-512	*bla* _KPC-3_	*bla*_TEM-1A_, *bla*_OXA-9_, *bla*_SHV-11_	0.12	0.06	≤0.007	0.015	0.015	0.56	**0.12**	**0.19**	**0.19**
KPB07 ^[a]^	*K. pneumoniae*	ST-1519	*bla* _KPC-3_	*bla*_TEM-1_, *bla*_OXA-9_, *bla*_SHV-11_	0.25	0.25	0.015	0.06	0.06	1.06	**0.18**	**0.30**	**0.30**
KPB09 ^[a]^	*K. pneumoniae*	ST-1519	*bla* _KPC-3_	*bla*_TEM-1_, *bla*_OXA-9_, *bla*_SHV-11_	0.125	0.125	≤0.007	0.015	0.015	1.06	**0.18**	**0.18**	**0.18**
KPB013 ^[a]^	*K. pneumoniae*	ST-1519	*bla* _KPC-3_	*bla*_OXA-9_, *bla*_SHV-11_	0.125	0.125	≤0.007	0.015	≤0.007	1.06	**0.18**	**0.18**	**0.12**
KPB02 ^[a]^	*K. pneumoniae*	ST-1519	*bla* _KPC-36_	*bla*_TEM-1_, *bla*_OXA-9_, *bla*_SHV-11_	0.25	0.25	≤0.007	0.06	0.06	1.06	**0.28**	**0.30**	**0.30**
KPC_TO3 ^[c]^	*K. pneumoniae*	ST-512	*bla* _KPC-66_	*bla*_TEM-1A_, *bla*_OXA-9_, *bla*_SHV-11_	0.5	≤0.007	≤0.007	0.015	0.015	**0.26**	**0.08**	**0.09**	**0.09**
BOT-EMOKP ^[a]^	*K. pneumoniae*	ST-1519	*bla*_KPC-31_, *bla*_KPC-3_	*bla*_TEM-1_, *bla*_OXA-9_, *bla*_SHV-11_	8	2	0.5	0.25	0.5	**0.31**	**0.12**	**0.09**	**0.12**
N435 ^[b]^	*K. pneumoniae*	-	*bla* _KPC-41_	*bla*_SHV-11_, *bla*_TEM-1_	4	0.5	0.12	0.12	0.5	**0.13**	**0.09**	**0.09**	**0.19**
N859 ^[b]^	*K. pneumoniae*	ST-258	*bla* _KPC-50_	*bla* _SHV-11_	16	16	4	1	4	1.06	**0.31**	**0.12**	**0.31**
CAZ59BO ^[a]^	*K. pneumoniae*	ST-512	*bla* _KPC-53_	*bla*_TEM-1_, *bla*_SHV-11_	2	0.5	0.125	0.125	0.125	**0.31**	**0.13**	**0.13**	**0.13**
R90 ^[b]^	*P. aeruginosa*	-	*bla* _KPC-2_	*bla* _TEM-1_	0.25	0.25	0.03	0.06	0.06	1.06	**0.18**	**0.30**	**0.30**
Ambler class B													
N590 ^[b]^	*E. coli*	ST-167	*bla* _NDM-5_	*bla* _CMY-42_	4	4	2	4	2	1.06	0.56	1.06	0.56
N1700 ^[b]^	*E. coli*	ST-69	*bla* _NDM-1_	*bla*_CMY-4_, *bla*_CTX-M-15_, *bla*_OXA-10_, *bla*_TEM-1B_	2	0.5	0.06	1	1	**0.31**	**0.28**	0.56	0.56
N2352 ^[b]^	*E. coli*	-	*bla* _NDM-5_	*bla*_CTX-M-15_, *bla*_OXA-1_, *bla*_TEM-190_	0.5	0.25	0.25	0.25	0.5	0.56	0.62	0.56	1.06
R2752 ^[b]^	*E.coli*	-	*bla* _VIM-34_	*bla* _TEM-1_	0.06	0.03	0.03	0.06	0.03	0.56	0.75	1.06	0.56
N1491 ^[b]^	*E. cloacae*	ST-78	*bla* _NDM-1_	*bla*_ACT-24_, *bla*_CTX-M-15_, *bla*_TEM-1_, *bla*_OXA-1_	4	4	2	4	4	1.06	0.62	1.06	1.06
N1692 ^[b]^	*K. pneumoniae*	ST-147	*bla* _NDM-1_	*bla*_CTX-M-15_, *bla*_OXA-140_, *bla*_OXA-9_, *bla*_SHV-11_, *bla*_TEM-1A_	8	4	2	4	4	0.56	**0.31**	0.56	0.56
N1697 ^[b]^	*C. freundi*	-	*bla* _NDM-1_	*bla*_OXA-1_, *bla*_SHV-12_	2	2	1	1	1	0.62	0.56	0.56	0.56
N1215 ^[b]^	*P. aeruginosa*	-	*bla* _VIM-2_	*bla*_OXA-486_, *bla*_PDC-3_, *bla*_PER-1_, *bla*_OXA-4_	0.25	0.125	0.125	0.25	0.06	0.56	0.56	1.06	**0.30**
N1539 ^[b]^	*P. aeruginosa*	ST-235	*bla* _NDM-1_	*bla*_PAO_, *bla*_OXA-50_	0.06	0.06	0.06	0.06	0.06	1.06	1.06	1.06	1.06
N1244 ^[b]^	*P. aeruginosa*	ST-111	*bla* _IMP-18_	*bla*_PDC-3_, *bla*_OXA-2_, *bla*_OXA-50_	0.03	0.015	0.015	0.03	0.03	0.56	0.56	1.06	1.06
N1744 ^[b]^	*P. aeruginosa*	ST-2613	*bla* _NDM-1_	*bla*_OXA-488_, *bla*_PAO-like_	1	1	0.5	0.5	0.5	1.06	0.56	0.56	0.56
Ambler class B+D													
N1898 ^[b]^	*A. baumannii*	ST-52	*bla*_NDM-9_, *bla*_OXA-58_	*bla*_ADC-158_, *bla*_OXA-98_	16	8	16	8	16	0.56	1.06	0.56	1.06
N2004 ^[b]^	*A. baumannii*	-	*bla*_NDM-1_, *bla*_OXA-23_	*bla*_OXA-66_, *bla*_ADC-30_	8	8	8	8	8	1.06	1.06	1.06	1.06
Ambler class D													
N1067 ^[b]^	*E. coli*	ST-38	*bla* _OXA-181_	*bla* _CTX-M-27_	0.25	0.03	0.015	0.06	0.03	**0.18**	**0.12**	**0.30**	**0.18**
N1085 ^[b]^	*E. coli*	ST-38	*bla* _OXA-244_	*bla* _CTX-M-27_	0.12	0.015	0.03	0.03	0.015	**0.19**	**0.37**	**0.31**	**0.19**
N1091 ^[b]^	*K. pneumoniae*	ST-11	*bla* _OXA-48_	*bla*_CTX-M-15_, *bla*_SHV-11_	0.5	0.06	0.015	0.12	0.06	**0.09**	**0.18**	**0.30**	**0.18**
N612 ^[b]^	*A. baumannii*	ST-2	*bla* _OXA-23_	*bla*_ADC-25like_, *bla*_OXA-66_	0.25	0.12	0.12	0.12	0.25	0.56	0.56	0.56	1.06
N774 ^[b]^	*A. baumannii*	ST-2	*bla* _OXA-40_	*bla*_ADC-25like_, *bla*_OXA-66_	1	0.5	0.5	1	0.5	0.56	0.62	0.56	1.06
N1183 ^[b]^	*A. baumannii*	ST-2	*bla* _OXA-23_	*bla*_ADC-25-like_, *bla*_OXA-66_	0.25	0.12	0.12	0.25	0.12	0.56	0.56	1.06	1.06
ACBB0432 ^[a]^	*A. baumannii*	ST-195	*bla* _OXA-23_	*bla* _TEM-1_	0.25	0.25	0.125	0.125	0.25	1.06	0.56	0.56	1.06
BO415CRAB ^[a]^	*A. baumannii*	ST-195	*bla* _OXA-23_	*bla* _TEM-1_	0.12	0.12	0.06	0.12	0.12	1.06	0.56	1.06	1.06
Reference													
ATCC 25922	*E. coli*	-	-	-	0.06	0.06	0.06	0.06	0.06	1.06	1.25	1.06	1.06
ATCC 700603	*K. pneumoniae*	-	-	*bla* _SHV-18_	0.25	0.015	0.06	0.06	0.03	**0.12**	**0.30**	**0.30**	**0.18**
ATCC BAA-2814	*K. pneumoniae*	-	*bla* _KPC_	*bla*_SHV-11_, *bla*_TEM-1_	1	0.03	0.5	0.06	0.06	**0.15**	0.56	**0.12**	**0.12**
ATCC 27853	*P. aeruginosa*	-	-	-	0.06	0.06	0.06	0.06	0.03	1.06	1.06	1.06	0.56

^[a]^ Operative Unit of Microbiology, IRCCS Azienda Ospedaliero-Universitaria di Bologna, Italy; ^[b]^ National Reference Center for Emerging Antibiotic Resistance, Switzerland; and ^[c]^ Microbiology and Virology Unit, University Hospital Città della Salute e della Scienza di Torino, Italy. Grey shading indicates resistance. Underline and bold character indicate additive and synergistic effect, respectively. Abbreviations: CFDC, cefiderocol; TAZ, tazobactam; AVI, avibactam; VAB, vaborbactam; REL, relebactam; and FICI, Fractional inhibitory concentration index.

## Data Availability

Not applicable.

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
