# Peer review of "Synergistic Effect of Clinically Available Beta-Lactamase Inhibitors Combined with Cefiderocol against Carbapenemase-Producing Gram-Negative Organisms"

_antibiotics, 2022, doi:10.3390/antibiotics11121681_

Round 1

Reviewer 1 Report

1. The introduction needs improvement.  

2. It was also required that the Sub-minimal inhibitory concentration (Sub-MIC at 0.5xMIC and 0.25xMIC) for each strain will againt each original MIC value must be calculated and documented. 

3. The details of data in table 1 are missing.   It is necessary to submit the supplementary file mentioning the original source of each classified strain in detail. 

4. Table 1 is the main source of synergistic effect evaluation, the author is requested to submit the original source of each and every strain their carbapenems gene details, and other beta-lactamase genes details to fulfill the COPD guidelines. 

Author Response

Referee: 1

Referee: 1

  1. The introduction needs improvement.

We thank the referee for this suggestion. Accordingly, we improved Introduction section.

  1. It was also required that the Sub-minimal inhibitory concentration (Sub-MIC at 0.5xMIC and 0.25xMIC) for each strain will againt each original MIC value must be calculated and documented.

We thank the referee for this comment. In this study we performed time kill assays using 1 X MIC of cefiderocol. Based on pharmacokinetic reasoning, we considered these concentrations more appropriate for the evaluation of synergy by time-kill assays. Further studies will be performed to assess the variable sub-MIC concentrations of cefiderocol.

  1. The details of data in table 1 are missing. It is necessary to submit the supplementary file mentioning the original source of each classified strain in detail.

We thank the referee for this comment. The strains used in the study are previously characterised. Most of the sequences of the strains are deposited on GenBank database and the remaining are in the process of being deposited. The main characteristics of the strains (sequence typing and β-lactamase genes content) are reported in Table 1. However, the authors confirm that all the strains used in this study are available from the corresponding author on reasonable request.

  1. Table 1 is the main source of synergistic effect evaluation, the author is requested to submit the original source of each and every strain their carbapenems gene details, and other beta-lactamase genes details to fulfill the COPD guidelines.

We thank the referee for this comment. The strains used in the study are previously characterised. Most of the sequences of the strains are deposited on GenBank database and the remaining are in the process of being deposited. The main characteristics of the strains (sequence typing and β-lactamase genes content) are reported in Table 1. However, the authors confirm that all the strains used in this study are available from the corresponding author on reasonable request.

Reviewer 2 Report

Review 

Title change to “Beta-lactamase inhibitors”

Make abstract structured 

Make subsections in the results and method sections 

Author Response

Referee: 2

  1. Title change to “Beta-lactamase inhibitors”

 We thank the referee for this comment. Accordingly, we modified the title.

  1. Make abstract structured

We thank the referee for this comment. In accordance with Journal instructions, the abstract should be a single paragraph and should follow the style of structured abstracts, but without headings.

  1. Make subsections in the results and method sections 

We thank the referee for this comment. Accordingly, we added subsections in both results and method sections.

Reviewer 3 Report

Dear Authors,

The manuscript presents a theme of significant scientific relevance. However, I have some comments.

  1. The terms in vitro and K. pneumoniae were not in the italic format during the manuscript;
  2. I suggest improving the introduction about antimicrobial resistance.

Author Response

Referee: 3

The manuscript presents a theme of significant scientific relevance. However, I have some comments.

We thank the referee for this appraisal.

  1. The terms in vitro and K. pneumoniae were not in the italic format during the manuscript.

We thank the referee for this comment. Accordingly, we modified the text.

  1. I suggest improving the introduction about antimicrobial resistance.

We thank the referee for this suggestion. Accordingly, we improved Introduction section.

Round 2

Reviewer 1 Report

The reply of author satisfy the reviewers comments 

Reviewer 2 Report

None

Reviewer 3 Report

Dear Authors,

I do not have any suggestions for the manuscript.

Best regards.